# The Use of Wearable Technologies in the Assessment of Physical Activity in Preschool- and School-Age Youth: Systematic Review

**DOI:** 10.3390/ijerph20043402

**Published:** 2023-02-15

**Authors:** António C. Sousa, Susana N. Ferrinho, Bruno Travassos

**Affiliations:** 1Research Center in Sports Sciences, Health Sciences and Human Development, CIDESD, 5001-801 Vila Real, Portugal; 2Department of Sport Sciences, University of Beira Interior, 6201-001 Covilhã, Portugal; 3Department of Letters, University of Beira Interior, 6201-001 Covilhã, Portugal

**Keywords:** wearable device, physical activity, assessment, children, preschool, school

## Abstract

In recent years, physical activity assessment has increasingly relied on wearable monitors to provide measures for surveillance, intervention, and epidemiological research. This present systematic review aimed to examine the current research about the utilization of wearable technology in the evaluation in physical activities of preschool- and school-age children. A database search (Web of Science, PubMed and Scopus) for original research articles was performed. A total of twenty-one articles met the inclusion criteria, and the Cochrane risk of bias tool was used. Wearable technology can actually be a very important instrument/tool to detect the movements and monitor the physical activity of children and adolescents. The results revealed that there are a few studies on the influence of these technologies on physical activity in schools, and most of them are descriptive. In line with previous research, the wearable devices can be used as a motivational tool to improve PA behaviors and in the evaluation of PA interventions. However, the different reliability levels of the different devices used in the studies can compromise the analysis and understanding of the results.

## 1. Introduction

In youth populations, effective health interventions contribute significantly to preventing obesity and metabolic diseases throughout life, as well as to the creation of an intention to practice such activities for life [1,2,3,4]. According to Sallis and Owen [5], it seems that children and adolescents are adopting the sedentary habits of adults, as well as their way of looking at physical exercise, namely, the usual reasons for not doing it. As adults adopt less active lifestyles and serve as role models for young people, it is natural for this to happen [6]. This situation calls attention to the need to first change the habits of adults, so that it is easier to intervene with younger people [6,7]. One of the factors that contributes to the sedentary lifestyle of young people is the reduction of physical efforts while they are commuting to school and in hobbies, namely watching television, playing electronic and computer games, socializing while sitting, etc. During their daily lives, they do not perform physical activity in sufficient amounts and intensities to promote beneficial effects on their health, namely in the prevention of risk factors [5]. Regardless of age, physical activity (PA) can be considered one of the most useful and successful strategies to promote health. PA is usually oriented towards daily habits that promote a healthy lifestyle or even to achieve optimal performance. For this reason, physical exercise improves the functioning of all of the systems of the human body, mainly the cardiovascular, respiratory, endocrine and digestive systems, strengthening the musculoskeletal system and increasing certain levels of flexibility [8].

To achieve such goals, PA assessment has been considered as a key factor to promote, monitor and also encourage practice among youths. PA assessment is increasingly dependent on wearable devices, as they are the excellent means to provide measures for evaluate surveillance, intervention and epidemiological research [9]. In recent years, there has been a sharp increase in the number and type of equipment available on the market to assess PA [9,10,11]. 

Wearable technologies are devices with sensors, screen, processor, memory and software with algorithms to filter, interpret, organize and store the raw data that is collected, which can connect to the internet and present the data to the user in real time or as a retrospective report [12]. In other studies, wearable technologies are defined as devices that can monitor the physical activities (steps, calories, etc.) [13,14], physiological data (heart rate, body temperature, etc.) [15,16,17], data [18,19,20], gesture detection [21,22] and emotion recognition [23] of the user 24/7 and collect and store and transmit these data, while helping the users to perform many other useful micro-tasks, such as checking incoming text messages and displaying urgent information [24,25,26].

Wearables serve as knowledge and risk management tools, becoming valuable business and sports management information technology tools. If the data they collect are properly read and strategic actions are drawn from them, their impact on sports performance could grow and become remarkable [27]. According to Çiçek [28], there are three main categories of wearable technology. These categories are health-related wearable technologies, textile-based wearable technologies, and consumer electronics wearable technologies. The benefits of wearable technologies are attested by the fact that they have been used for a long time in various disciplines and for different purposes, such as medical sciences, fashion, and sports [29]. In fact, wearables are very popular these days and have great potential due to the fact that they can improve people’s daily lives, and it is accepted that the application of sensors and other electronic devices on the body can revolutionize the human experience in several areas [29]. 

In addition, these devices have some advantages, one of which is that they can be easily used in any field environment (increasing ecological validity) so that coaches/teachers can obtain feedback (positive or negative) on the variables measured in real time and report the performance level of players/athletes/students, whether during training, a game or a class [30]. The data obtained through these devices/sensors can be used for a variety of purposes based on the objectives outlined at the beginning of each study, such as: (i) measuring, controlling and increasing the physical performance of players/athletes/students; (ii) preventing possible injuries caused by excessive overload (in training or game); (iii) preventing injured players/athletes returning prematurely to training/a game; (iv) monitoring and predicting the performance evolution of younger players/athletes. In addition, these devices/sensors are programmed/manufactured to operate in any sports environment (outdoor or indoor venues) as they are small, light, wireless and easy to transport [30]. Finally, some devices/sensors have additional features, such as being waterproof or having the ability to store data at low temperatures [31,32,33].

Research on the use of wearable devices in the school context of physical education has shown that this type of application brings advantages in terms of motivation, knowledge of the results, evaluation and improvement of the student’s autonomy [34,35,36,37,38]. Schwartz and Baca [39] point out that most of these AP applications are based on behavioral theory and use gamification elements to achieve success with personal goals and specific feedback. Therefore, Lee and Gao [40] recommend the use of apps to particularly facilitate students’ group activities, as well as understanding the impact of such practices on the results.

However, little is still known about the use of these wearable technologies in schools. Therefore, our purpose was to examine the current research about the use of wearable technology in the evaluation in physical activities of preschool- and school-age children and to provide teachers/researchers with perspectives for future lines of research.

## 2. Materials and Methods

This review was conducted using the Preferred Reporting Items for Systematic Reviews and Meta-Analysis (PRISMA) statement guidelines [41]. 

### 2.1. Search Strategy 

The search was performed across the entire literature between 2000 and August 2022 in three electronic databases: Web of Science, PubMed and Scopus. The Boolean search method was used, which limited the search results with operators including AND/OR only to studies that contained key terms relevant to the scope of this review. The search terms were identified: “children” OR “adolescents” AND “wearable” OR “portable sensors” OR “sensors” OR “accelerometers” AND “physical activity” OR “physical exercise” OR “endurance” OR “aerobic” OR “strength training” OR “resistance training” AND “monitoring” AND “school” OR “physical education”. Appendix A Appendix A reports the search strategies used in the three databases.

### 2.2. Eligibility Criteria 

The included studies focused on wearable technologies in the preschool and school context in youth with physical activity-related outcomes. Studies published in English in a peer-reviewed journal were included, evaluating physical activity in all school settings (playtime, physical education classes and school sports) in healthy, young people. Review articles (i.e., qualitative reviews, systematic reviews and meta-analyses), theses, dissertations, congress abstracts and proceedings were not considered. All of the information collected from the studies included in the systematic review were based on the research design, objective, subjects, procedures and findings.

### 2.3. Study Selection

The systematic search identified 474 records. After an initial screening, 29 studies were considered to be eligible for evaluation, and those that did not meet the inclusion criteria were excluded (e.g., inconclusive information about study procedures, the intervention method, etc.). In the end, a total of 21 studies were included in the final qualitative analysis. The earliest of these studies was published in 2001 [42], and the most recently published study was from 2020 [43]. Figure 1 shows the article selection process.

### 2.4. Data Extraction and Synthesis 

The included articles obtained information on sample size, age, wearable strategies, measurements, main results and conclusions. The data extraction process was performed by two authors (A.C.S. and B.T.), and inconsistent data were resolved by the third author (S.N.F.). 

### 2.5. Data Analysis

#### Assessment Risk of Bias

To assess the risk of bias, the Cochrane Reviews method was used [44]. Two authors (A.C.S. and S.N.F.) assessed the risk of bias of each study against key criteria: random sequence generation, allocation concealment, blinding of outcome assessment, blinding participants and personnel, incomplete outcome data, selective reporting and other bias, and when there was no consensus between the two authors on which classification to assign to a given criterion, a third author evaluated the study (B.T.). In classifying the studies, the following terms were used: low risk, high risk or unclear risk. The Review Manager software (RevMan, The Nordic Cochrane Centre, Copenhagen, Denmark) version 5.4 was used to build the risk of bias graphs.

## 3. Results

### 3.1. Description of the Studies Reviewed

Table 1 presents all of the information about the studies included in the review. The studies included sample sizes from 19 to 1908 subjects (boys and girls) aged from 3 to 19 years old. The sample came from thirteen different countries: five studies in the USA [45,46,47,48,49], four studies in the Czech Republic [43,50,51,52], two studies each in Australia [53,54], Poland [50,55] and England [56,57], and one study each in Estonia [58], Mexico [49], Portugal [59], Norway [60], New Zealand [61], France [42], Sweden [62] and the Netherlands [63]. 

Of the 21 studies reviewed on physical activity in the preschool and school context, 71.5% (*n* = 15) developed an intervention for distance performance, 9.5% (*n* = 2) applied an intervention for the number of steps and 19% (*n* = 4) focused on both of the interventions mentioned above. Regarding the wearable technology used in the studies, 80.1% (*n* = 17) of the studies focused on accelerometers, while 24% (*n* = 6) were based on pedometers. In most of the studies, the participants wore the device on their hip or near their center of gravity (80%) (*n* = 19), and only in 10% (*n* = 2) of the participants wore the device on their arm/wrist. 

Table 2 shows the main characteristics of the devices used in the studies that were included in this review. Thirteen different devices were registered for the development of 21 studies with different methods of registration and data analysis (for detailed information, see Table 2). 

### 3.2. Risk of Bias in the Included Articles

About 30.0% of the studies were randomized, and 70.0% used cross-sectional designs. The generated allocation sequence item identified the least often applied item, but it does not provide sufficient detail to assess whether it could produce comparable groups. Most investigations implemented a blinded design; however, a few studies performed a cross-group comparison. About 60.0% of the studies revealed their concealed allocation, which had a systematic bias of therapeutic effectiveness, and 90.0% of the studies reported a low risk of bias in the incomplete outcome data (attrition bias domain), which revealed transparency in the methodology used. Well-reported losses and exclusions were reported in the studies [44] (Figure 2 and Figure 3).

### 3.3. Physical Activity in Children’s Using Pedometers

In the articles included in this review, the only ones that used pedometers exclusively were the studies by Mitas et al. [51] and Scruggs et al. [47] (Table 3). The study by Mitas et al. [51] found that boys spent more time performing vigorous physical activity (VPA) and moderate physical activity (MPA) than girls did. However, in the study by Scruggs et al. [47], it was observed that first grade girls (6 to 10 years) spent more time performing moderate–vigorous physical activity (MVPA) and VPA than boys did. In the second year, it was found that girls spent more time performing MVPA than boys did, while the latter group spent more time performing VPA, concerning the validation sample group. In the cross-validation group, the first grade boys (6 to 10 years) spent more time performing MVPA and VPA, while second grade boys (10 to 14 years) spent the same amount of time doing this as first grade boys and girls did (6 to 10 years).

Therefore, in line with the results, boys generally are more predisposed to practicing more PA, whether it is moderate or vigorous, than girls are during their daily routines at school.

### 3.4. Physical Activity in Children’s Using Accelerometers

In general, the studies that included accelerometer devices revealed that most of the children in the analyses spent most of their time performing MVPA [45,46,53,55,56,58,59,63]. However, in opposition, there were two studies [50,60] that revealed that children tend to spent most of the time performing light physical activity (LPA) (Table 3).

However, five studies were included in the review with reports of the significant effects on the PA performed [42,43,48,49,57]. For example, in the study by Blaes et al. [42], significant differences were observed between the PS vs. JHS groups for LPA (*p* < 0.05), for MPA, VPA and very high physical activity (VHPA) levels in the PS vs. Ps and PS vs. JHS groups (*p* < 0.05) and for the PS vs. JHS group in the moderate–very high physical activity (MVHPA) (*p* < 0.05).

The comparison between normal and obese children in the study by Gao et al. [49] revealed, as expected, that normal children spent more time performing physical activity (percentage of MVPA) than obese children do (*p* = 0.029, η2 = 0.03) However, it was found that obese children spent more time being sedentary than normal children do (*p* = 0.002, η2 = 0.06).

The comparison between boys and girls revealed that, in general, boys are more predisposed to and practice more PA at a high intensity than girls do [43,48,57]. Results of Kerr et al. [57] reported that girls spend more time performing LPA than boys do (*p* < 0.01, d = 1.21). In contrast, boys spend more time performing VPA (*p* < 0.01, d = −1.04) and VHPA (*p* < 0.01, d = −0.82), respectively, than girls do. In the study by Saint-Maurice et al. [48], boys were significantly more active than girls were in different schools (*p* < 0.001). Finally, in the study by Valach et al. [43], boys revealed significantly high BP levels during recess while performing LPA (*p* < 0.01, η2 = 0.032) and VPA (*p* < 0.01, η2 = 0.061) and VPA at school (*p* < 0.001, η2 = 0.050) than girls did.

### 3.5. Physical Activity in Children’s Pedometers vs. Accelerometers

Only four studies [52,54,61,62] used pedometers and accelerometers simultaneously in their analyses (Table 3). In the study by Hartwig et al. [54], it was observed that the physical activity assessment was more effective for pedometers in the three experimental groups than it was for accelerometers. Additionally, in line with previous results, boys tend to present higher values of PA than girls did.

In the study by Sigmund et al. [52], there were significant effects caused by accelerometers (*p* < 0.001, d = 1.5646) and pedometers (*p* < 0.001, d = 1.3231) on general physical activity. Furthermore, both devices found significant effects (*p* < 0.001) when the authors were comparing the differences between the boys and girls. In the study by Rush et al. [61], boys and girls have higher values for physical inactivity than they do for physical activity throughout a school day. Finally, in the study by Raustorp et al. [62], it was observed that boys spent more time performing MVPA than girls did during a physical education class.

## 4. Discussion

This review aimed to examine the current research about the use of wearable technology in the evaluation in physical activities of preschool- and school-age children and to provide teachers/researchers with perspectives for future research. Studies on this topic are relatively new, with a higher incidence in recent years. However, only twenty-one studies have been observed to assess and understand PA in schools using wearable technology in last 22 years. In general, the studies were descriptive and compared the practice of boys and girls or even measured the reliability of the devices. Most of the studies used accelerometers, and only four of them used pedometers. However, thirteen different devices were registered for the development of the studies with different methods of registration and data analysis, creating difficulties to further understand the tendencies of results and define recommendations for the future. In general, and in line with previous research, wearable technology can be used as a motivational tool to improve PA behaviors and in the evaluation of PA interventions [40,41].

Many studies have evaluated the accuracy of various pedometers. Pedometers are usually simple and inexpensive devices, giving real-time feedback in terms of measuring the number of steps taken on a daily basis [42,43,44,45]. The pedometers revealed low accuracy at slower speeds, particularly the ones that used a spring-suspended horizontal lever arm mechanism [43,45]. In addition, pedometers may have low accuracy when they are attached to other parts of the body [46] or when they are attached to certain clothing items (e.g., when wearing a dress) [47]. These issues are fundamental for gaining an understanding about the obtained results and their application to the practice and for the development of more adjusted programs of intervention. However, despite its importance to improving the user acceptance of pedometers, to date, a small amount of research had been developed to explore the reliability of the pedometers and to validate their use in different contexts and types of activities [64].

In addition to pedometers, the use of accelerometers has also increased in recent years when one has been assessing PA [48]. Triaxial accelerometers are motion devices that measure acceleration in three planes during body motions [49]. Therefore, they were developed to measure PA levels and provide information that motivates individuals to exercise. Compared to pedometers, accelerometers are the devices that are most often used by researchers and in clinical settings because they have more variables that can be analyzed. For example, while pedometers only assess the distance covered by the number of steps, accelerometers allow us to assess the frequency, duration and intensity of PA [49]. Both of the devices showed good validity in terms of activity count (number of steps) and energy expenditure in different populations (healthy and chronically ill populations) [50,51].

The use of wearable technology in a preschool and school contexts has been studied since the beginning of the 1980s. However, researchers have only recently focused on realizing PA’s preponderance in children, whether they are in the classroom or the playground. All of the studies implemented interventions based on wearable technologies (using accelerometers and pedometers) with a period between two days and ten months in terms of PA performed in PE classes or during recess, and the authors found positive effects of the intervention on the different intensities used by children during PA. The included studies revealed that boys spend more time performing PA than girls do. Another essential result to highlight is that, in general, they (boys and girls) spend more time performing MVPA compared to LPA and VPA.

PA is a very important and essential tool for the adoption of healthy habits in children, but also for their future life, as several studies have shown a positive association between PA and gender [52,53], whether in the form of leisure [54,55] or in the form of physical activity in education/recreation classes [56,57]. On the other hand, there are some studies that have not reported a positive relationship between PA and gender [58,59,60,61].

The overall differences between PE days and non-PE days indicate that an additional 19 min of high-intensity physical activity (vigorous physical activity and above) during the PE day is critical, as vigorous physical activity (or greater) is a stronger predictor of cardiorespiratory fitness [65,66,67,68], body fat [69,70,71] and vascular function [72] in children compared with that of moderate–intensity physical activity. Thus, the daily use of pedometers, but particularly accelerometers, is highly recommended in schools to help PE teachers to classify the level of the students’ activity and the adequate application of training loads to different groups. For example, while the practice of team sports should be encouraged as an efficient way to promote higher levels of PA in some students, others could be encouraged to practice different activities with a low PA impact, but to develop coordination or other kind of capacities. The use of these devices to promote the gamification of the practice of PA should be encouraged and promoted using dedicated apps for mobile phones or tablets. A lot of systematic reviews have been published in recent years related to the subject of this review, where the authors [73] found that the PA level is also influenced by the students’ friends’ PA level, demonstrating that team sports and community life can increase the PA level, promoting the reduction of the obesity-related problems. Another study also verified that using Fitbit devices may benefit increasing the level of PA during recreational activities [74]. The mobile applications for this subject are more related to the measurements of weight, height, age, gender, goals, and calories needed for calculations, diet diaries and food databases including calories, calories burned and calorie intake [75].

This systematic review presents some limitations that must be recognized: (i) most of the articles included were based solely and exclusively on pedometers and accelerometers for the analysis of the children’s PA in preschool and school contexts; (ii) only two articles included in this review focused on energy expenditure (kcal) during PA performed at school and (iii) the risk of bias of the included studies reported that a third of the studies did not describe how the distribution of the groups was carried out.

Future research on PA time trends in schools could use a similar strategy using a subjective assessment of PA with the subsequent objective monitoring of PA. It also seems appropriate to use the newly verified Youth Activity Profile [76] and to monitor, at least weekly, PA using simple wearable devices (wristbands) that are suitable for longitudinal use. Finally, strategies that promote participant adherence to the monitoring protocol should be emphasized [77]. Therefore, future investigations should not focus only on general data, but they should also seek to discriminate the contexts or activities associated with the practice of physical activity in order to better understand what distinguishes boys and girls.

## 5. Conclusions

Wearable technology can be an essential instrument/tool to detect movements and monitor the physical activity of children and adolescents. There are known benefits of using these instruments to measure the levels of physical activity and the daily energetic cost (accelerometers: ActiTrainer, ActiGraph GT3X and Fitbit Zip and Fitbit Flex; pedometers: Yamax Digiwalker SW200 a Yamax Digiwalker SW700). In terms of the reliability and validity of pedometers and accelerometers and based on currently available evidence, we conclude that the ActiGraph accelerometers (in particular, the GT3X versions), Actical and ActiTrainer, have the best measurement properties to assess common movement-related outcomes (e.g., example, MVPA and TPA) for school-based activities for preschool- and school-aged children, and they should be the tools of choice where resources permit it is and where it is logistically possible. On the other hand, Fitbit Zip and Fitbit Flex also showed very promising results; however, these were based on a very limited sample of studies. On the other hand, we found that the Yamax Digi-Walker (SW-200) and Yamax Digi-Walker (SW-700 and 701) pedometers have the best measurement characteristics related to movement (e.g., example, MVPA and VPA). However, there are only a few studies on the influence of these technologies on physical activity in schools. Therefore, there was a large number of different devices and methods considered in the studies, which did not allow us to further understand the best practices or to define some recommendations for the future. In line with that, more studies with larger samples of the population involved and with more methods and procedures are required to really understand the effect of such programs on physical activity (e.g., the number of calories burned and the number of steps performed), as well as health. To improve upon the descriptive studies that only registered and compared the physical activity in school, future research should be focused on the use of such devices in specific intervention programs that evaluate different groups using the same devices and variables, which later, could be used to intervene by improving young people’s health and instilling healthy lifestyles.

## Figures and Tables

**Figure 1 ijerph-20-03402-f001:**
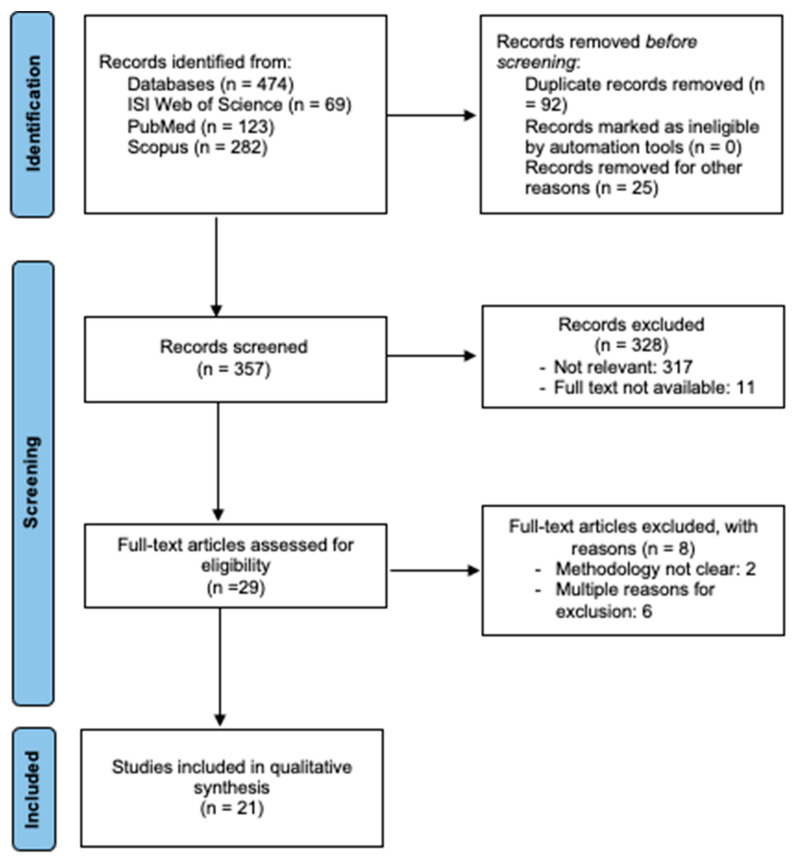
PRISMA study flow diagram.

**Figure 2 ijerph-20-03402-f002:**
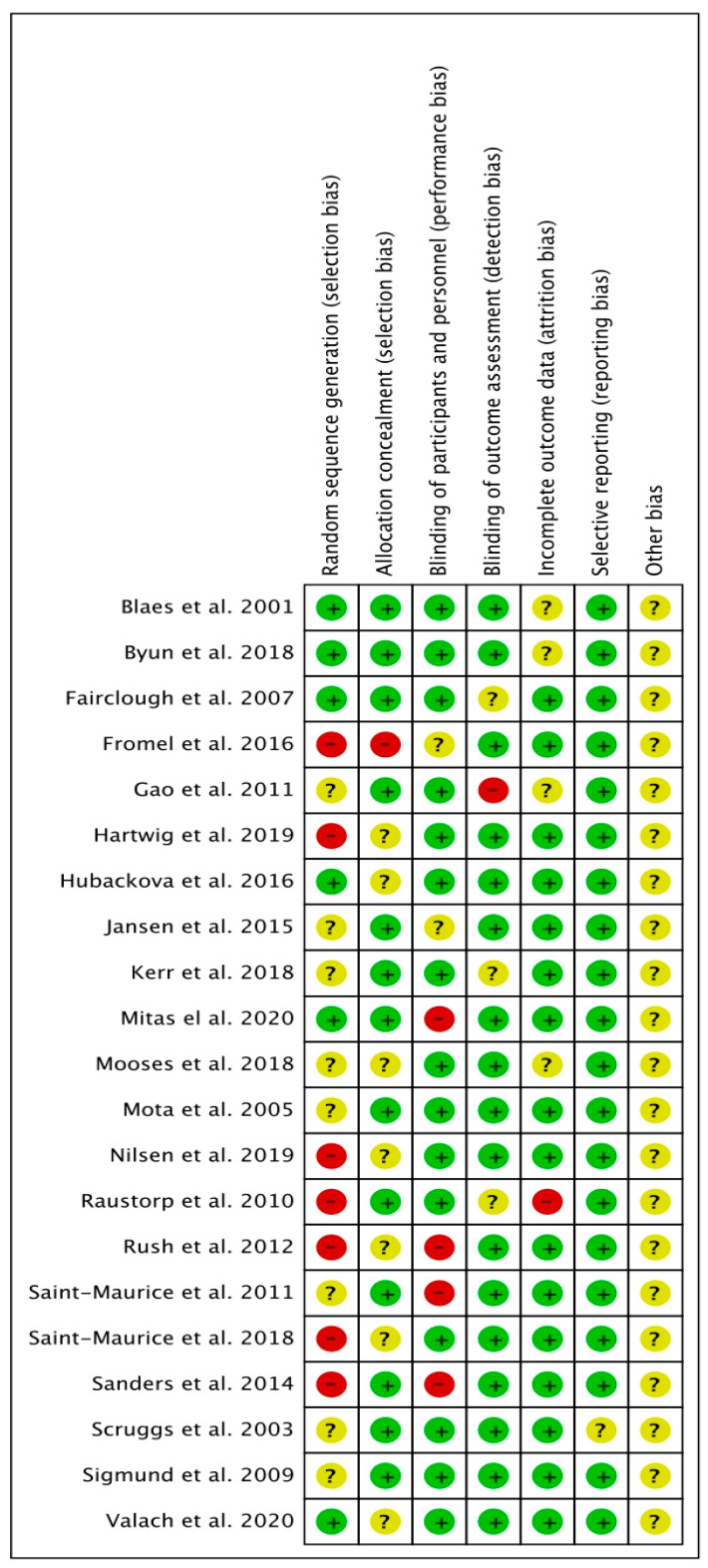
Judgments about each risk of bias item for each included study [42,43,45,46,47,48,49,50,51,52,53,54,55,56,57,58,59,60,61,62,63].

**Figure 3 ijerph-20-03402-f003:**
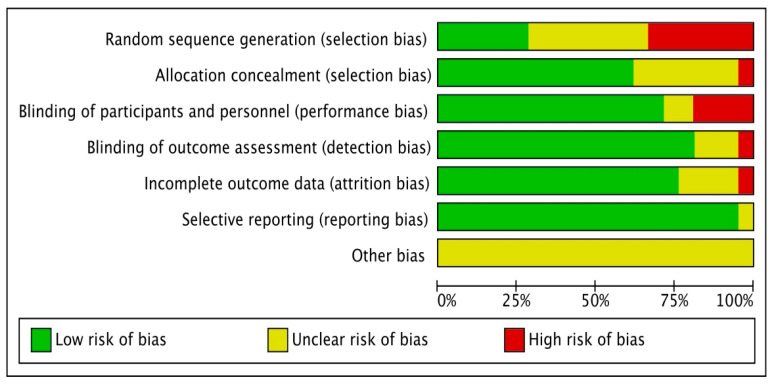
Risk of bias items are presented as percentages across all included studies.

**Table 1 ijerph-20-03402-t001:** Characteristics of the studies included in the review.

Authors/Year/ Country	Sample/Age (Years)	Device	Data Collection Procedures	Outocomes
Valach et al. [43] Czech Republic	*n* = 212 (136 girls and 76 boys) Age = 16–19 years	ActiTrainer^TM^ accelerometer	Participants wore the device on their right hip.	ActiTrainer^TM^ evaluated distance performed
Frömel et al. [50] Czech Republic and Poland	*n* = 333 (223 girls and 110 boys) Age = 15–18 years	ActiTrainer^TM^ accelerometer	Participants wore the device on their right hip.	ActiTrainer^TM^ evaluated distance performed
Hubáčková et al. [55] Poland	*n* = 395 (238 girls and 157 boys) Age = 9–17 years	ActiTrainer^TM^ accelerometer	Participants wore the device on their right hip.	ActiTrainer^TM^ evaluated distance performed
Mitáš et al. [51] Czech Republic	*n* = 1908 (1129 girls and 779 boys) Age = 15–19 years	Yamax Digiwalker SW-700 pedometer	The pedometer was placed on the hip in the region of the body’s center of gravity.	Yamax Digiwalker SW-700 evaluated the number of steps (total)
Byun et al. [45] USA	*n* = 93 girls Age = 4–5 years	Firbit Flex and ActiGraph GT3X+ accelerometers	The accelerometer was attached to the child’s hip using an elastic belt. Fitbit Flex were fitted to age-appropriate wristbands on the non-dominant wrist.	Firbit Flex and ActiGraph GT3X+ evaluated distance performed
Saint-Maurice et al. [46] USA	*n* = 291 (163 girls and 128 boys) Age = 8–17 years	SenseWear Armband	SenseWear Armband is worn on the back of the upper arm.	SenseWear Armband evaluated distance performed
Sanders et al. [53] Australia	*n* = 133 boys Age = 14 years	ActiGraph GT3X accelerometer	The accelerometer was placed on the upper part of the right hip.	ActiGraph GT3X evaluated distance performed
Scruggs et al. [47] USA	*n* = 369 (178 girls and 191 boys) Age = 7–8 years	Yamax DigiWalker SW-701 pedometer	Pedometer placement was standardized by placing them on the belt or waistband, approximately 5–7 cm from the umbilicus.	Yamax Digiwalker SW-701 evaluated the number of steps (total)
Mooses et al. [58] Estonia	*n* = 144 (72 girls and 72 boys) Age = 9–10 years	Fitbit Zip and an ActiGraph GT3x-accelerometer	The accelerometer and Fitbit Zip were attached on the hip with the same elastic belt and worn on the same side.	Both accelerometers were used to assess distance performed.
Kerr et al. [57] England	*n* = 36 girls Age = 11–12 years	RT3^@^ triaxial accelerometer	All of the children were asked to strap an accelerometer to their waist.	RT3^@^ trixial evaluated distance performed
Janssen et al. [63] Netherlands	*n* = 1486 (658 girls and 828 boys) Age = 7–10 years	ActiTrainer^TM^ and ActiGraph accelerometers	The accelerometer was securely attached to the children’s hip by an elastic waist belt.	ActiGraph and ActiTrainer^TM^ evaluated distance performed
Saint-Maurice et al. [48] USA	*n* = 100 (48 girls and 52 boys) Age = 8–12 years	ActiGraph GT1M accelerometer	Participants were asked to wear the monitor on the right side of their hip (at waist level) during the entire school day.	ActiGraph GT1M evaluated distance performed
Hartwig et al. [54] Australia	*n* = 592 girls Age =13–14 years	ActiGraph GT3X+ accelerometer and Yamax Digiwalker SW-200 pedometer	The participants used ActiGraph GT3X+ accelerometer and Yamax Digi-Walker SW pedometer on an elastic belt secured across their hips during PE lessons.	ActiGraph GT3X+ evaluated distance performed, while Yamax Digiwalker SW-200 evaluated the number of steps (total)
Sigmund et al. [52] Czech Republic	*n* = 176 (84 girls and 92 boys) Age = 5–7 years	Caltrac one-axial accelerometer and Yamax Digiwalker SW-200 pedometer	The Caltrac accelerometer and the Yamax pedometer were attached to elastic belts on the children’s right hips.	Caltrac evaluated energy expenditure (distance performed), while Yamax Digiwalker SW-200 evaluated the number of steps (total)
Gao et al. [40] USA and Mexico	*n* = 149 (74 girls and 75 boys) Age = 10–14 years	Actical Mini-Mitter accelerometer	The accelerometer was placed with a belt support on the left hip.	Actical Mini-Mitter evaluated distance performed
Mota et al. [59] Portugal	*n* = 22 (12 girls and 10 boys) Age = 8–10 years	CSA 7164 accelerometer	The accelerometer was placed in a small nylon pouch and firmly adjusted at the child’s waist by an elastic belt over the non-preferred hip.	CSA 7164 evaluated distance performed
Fairclough et al. [56] England	*n* = 58 (27 girls and 31 boys) Age = 7–11 years BMI = 19.5 girls and boys	ActiGraph GT1M accelerometer	Children were asked to wear the accelerometer ActiGraphs attached to their right hip.	ActiGraph GT1M evaluated distance performed
Nilsen et al. [60] Norway	*n* = 1109 (537 girls and 572 boys) Age = 4–5 years	ActiGraph GT3X+ accelerometer	The accelerometers were mounted on the right hip.	ActiGraph GT3X+ evaluated distance performed.
Rush et al. [61] New Zealand	*n* = 47 (28 girls and 19 boys) Age = 8–11 years	Yamax SW-200 pedometer and Actical Mini-Mitter accelerometer	The pedometer and accelerometer were attached to the belt. It was then sealed with a cable tie and positioned on the child who wore the same pedometer/accelerometer combination each day.	Actical Mini-Mitter evaluated distance performed, while Yamax Digiwalker SW-200 evaluated the number of steps (total)
Blaes et al. [42] France	*n* = 361 (193 girls and 168 boys) Age = 3–16 years	ActiGraph GT1M accelerometer	The accelerometer was placed on the right hip, secured with an elastic belt.	ActiGraph GT3X+ evaluated distance performed.
Raustorp et al. [62] Sweden	*n* = 19 (9 girls and 10 boys) Age = 10 years BMI = 16.9 girls and 18.3 boys)	Yamax SW-200 pedometer and ActiGraph GT1M accelerometer	The pedometer and accelerometer were attached to an elastic belt to the waistband and placed in line with the midpoint of the right knee.	ActiGraph GT1M evaluated distance performed, while Yamax Digiwalker SW-200 evaluated the number of steps (total)

**Table 2 ijerph-20-03402-t002:** Characteristics of the devices included in the review.

Device Technology	Characteristics
ActiTrainer^TM^ accelerometer	The ActiTrainer accelerometer is surrounded by a metal shield and packaged into a plastic enclosure measuring 50 × 40 × 15 mm, weighing 45 g and including a 3-V (2430) coin cell lithium battery, and it has a dynamic range of 0.25 to 2.5 g, a sampling frequency of 30 Hz and contains a cantilevered rectangular piezoelectric bimorph plate and seismic mass, a charge amplifier, analog band-pass filters and a voltage regulator to measure acceleration in a single axis. The filtered acceleration signals (in the longitudinal axis) generate counts the magnitude of which is summed over a user-specific time (an epoch interval).
Yamax Digiwalker SW-700/701 pedometer	Digiwalker SW-700/701 pedometer is a simple motion sensor (5.0 × 3.8 × 1.4 cm and 21 g) that has a relatively low cost and is worn on the waist, which provides the number of steps performed in a given period, in addition to distance traveled and energy expenditure. Its mechanism consists of a suspended spring system that oscillates according to the vertical movement of the hip. Each spring deflection is recorded as a step and using this measurement and the anthropometric data, the energy expenditure is estimated.
SenseWear Armband	SenseWear Pro2 Armband is a multiple-sensor device collecting data using a skin temperature sensor, near body temperature sensor, heat flux sensor, galvanic skin response sensor and a biaxial accelerometer. The skin temperature sensor and near-body temperature sensor (a vent on the side of the armband) consists of sensitive thermistors that are in contact with the skin, relying on changes in resistance with changing temperature. The heat flux sensor uses the difference between skin temperature and near-body temperature to assess the heat loss. The galvanic skin response sensor measures the conductivity of the skin between two electrodes that are in contact with the skin. The conductivity of the skin varies according to physical and emotional stimuli. The biaxial accelerometer registers the movement of the upper arm and provides information about body position. The information from the sensors, together with gender, age, height and weight, are incorporated into proprietary algorithms to estimate energy expenditure. These algorithms are activity specific and are automatically applied on the basis of an analysis of the pattern of signals from the sensors.
Actigraph GT3X+ accelerometer	The Actigraph GT3X+ accelerometer is a light (19 g) and small (4.6 × 3.3 × 1.5 cm) device that detects bodily movements using a triaxial accelerometer at a dynamic range of ±6 g. Users can initialize the Actigraph GT3X+ accelerometer with sampling frequencies from 30 to 100 Hz, and export data in 1 to 60 s epochs.
Actigraph GT3X accelerometer	The Actigraph GT3X monitor device is lightweight (27 g), compact (3.8 × 3.7 × 1.8 cm) and has a rechargeable lithium polymer battery. It uses a solid-state tri-axial accelerometer to collect motion data on 3 axes: vertical (Y), horizontal right–left (X) and horizontal front–back axes (Z). The Actigraph output also includes the VM. The GT3X measures and records time-varying accelerations ranging in magnitude from ~0.05 to 2.5 Gs. The accelerometer output is digitized by a 12-bit analog to digital converter (ADC) at a rate of 30 Hz. Once digitized, the signal passes through a digital filter that band-limits the accelerometer to the frequency range of 0.25–2.5 Hz.
Fitbit Zip	The Fitbit Zip is a triaxial accelerometer that can measure the number of steps taken, distance traveled, and calories burned. This monitor is small (35.6 × 28.9 × 9.6 mm) but has an expanded battery life—approximately 4–6 months—and is less expensive than most other accelerometers.
Firbit flex	The Fitbit Flex (FF) features a triaxial accelerometer, and it is light (14 g including wristband) and small (3.2 × 1.2 × 1.0 cm). Using Bluetooth technology, the recorded data are wirelessly transferred to a cloud-based Fitbit application program interface called Fitbit dashboard (Fitbit.com), in which users find their data such as steps, total energy expenditure, ambulatory distance and active minutes (corresponding to MVPA). The data collected by the FF can be downloaded to Microsoft Excel or comma-separated values files via the Fitbit dashboard; however, the resolution of data exported from the Fitbit dashboard is lower than those of the other accelerometers.
RT3^@^ triaxial accelerometer	The RT3 accelerometer is a small (71 × 56 × 28 mm), lightweight (65.2 g), battery-powered instrument used as an experimental tool for measuring the physical activity of people. It is worn clipped to the waistband as an “accessory” during waking hours. Depending on its mode of operation, it can record data for up to 21 d, which are then downloaded to a PC for display and statistical processing. The sensor in the RT3 is an accelerometer sensitive along three orthogonal axes (X, Y and Z), which represent vertical, anteroposterior and mediolateral motions, respectively. The acceleration is measured periodically, converted to a digital representation and processed to obtain an “activity count,” which is stored in memory. The exact relationship of the activity count to the acceleration (measured in meters per second squared or g, where 1 g = 9.81 m·s^−2^) is not clear. The RT3 has four modes of operation: mode 1 samples and stores the activity counts on individual axes at 1 s epochs; mode 2 samples and stores vector magnitude (a measure combining all three axes of motion) activity counts at 1 s epochs; mode 3 samples and stores accumulated activity counts on individual axes over 1 min epochs; mode 4 samples and stores accumulated vector magnitude activity counts over 1 min epochs.
Yamax Digiwalker SW-200 pedometer	The Yamax Digiwalker SW-200 is a non-expensive, small (5.0 × 3.8 × 1.4 cm) and light electronic pedometer (21 g). Using a pendulum arm moving with the vertical oscillations of walking, its circuit switches on and off. Each vertical oscillation that exceeds the device threshold (#0.35 g) counts as a step. The total step count, which is the most accurate pedometer-derived variable representing PA, is shown on the display of the device.
Actigraph GT1M accelerometer	The Actigraph GT1M (mass, 27 g; 3.8 × 3.7 × 1.8 cm) uses an omnidirectional accelerometer to sense vertical accelerations, which range between 0.05 and 2.0 Gs; however, in its latest version (V3), it is possible to obtain counts from two axes. The accelerometer output is digitized by a twelve-bit Analog to Digital Convertor (ADC) at a rate of 30 Hz.
Caltrac one-axial accelerometer	The Caltrac one-axial accelerometer (Muscle Dynamics Fitness Network, Torrance, CA, USA) is small and light (<80 g) and measures vertical movement. Total and activity energy expenditure is estimated by entering the participant’s age, height, weight and sex; cumulative energy expenditure values are displayed on a screen. The Caltrac functions are such that when the trunk accelerates, the accelerometer produces a charge that is proportional to the force exerted by the subject, generating an acceleration–deceleration wave. The area under this wave is summed to yield the final number value of AEE.
Actical Mini-Mitter accelerometer	The Actical accelerometer (Mini Mitter) has an omnidirectional sensor and is capable of measuring movement in one plane. The sensor functions via a cantilevered rectangular piezoelectric bimorph plate and seismic mass, and it is capable of detecting movements in the 0.5 to 3 Hz range. Voltage generated by the sensor is amplified and filtered via analog circuitry. The amplified and filtered voltage is passed into an analog to digital converter, and the process is repeated 32 times per second (32 Hz). The resulting 1 s value is divided by four, then added to an accumulated activity value for the epoch. The Actical is the smallest accelerometer available (28 × 27 × 10 mm, 17 g) and is water resistant.
CSA 7164 accelerometer	The uni-axial CSA accelerometer is a small (5.1 × 4.1 × 1.5 cm), lightweight (42.5 g), single-channel accelerometer designed to measure and record acceleration ranging in magnitude from 0.05 to 2.00 g with a frequency response from 0.25 to 2.50 Hz. The filtered acceleration signal is digitized, and the magnitude is summed over a user-specified period of time (an epoch interval). At the end of each epoch, the summed value is stored in memory, and the numerical integrator is reset. This process can repeat itself for 22 consecutive days if a 1 min epoch is used before the memory is filled. Using a reader interface unit connected to a computer it is possible to download the recorded data and, using the software supplied with the unit, analyse the data.

**Table 3 ijerph-20-03402-t003:** Analysis of the main results of the studies included in the review.

References	Main Aim	Main Findings
Valach et al. [43]	To investigate the differences in the volume and intensity of PA between girls and boys with different levels of academic achievement throughout of a school day	**PA during school**
		LPA (<3 METs (min·h^−1^)): H = 13.74, *p* < 0.01, η2 = 0.032 between boys and girls with better AA.
		VPA (>6 METs (min·h^−1^)): H = 26.27, *p* < 0.01; η2 = 0.061 between boys and girls with worse AA, with both of them in favor of boys.
		**PA recess**
		VPA (>6 METs (min·h^−1^): H = 21.58; *p* < 0.001; η2 = 0.050), where boys with better AA were physically more active than girls were with better AA.
Frömel et al. [50]	To examine the differences in the intensity of PA during school days and weekends in Polish and Czech boys and girls	**Intensity PA during school days (average)**
		<3 METs (hour·min^−1^) (LPA): Boys (Czech—9.77; Polish—9.48); Girls (Czech—8.68; Polish—8.56).
		3—5.9 METs (hour·min^−1^) (MPA): Boys (Czech—1.69; Polish—1.33); Girls (Czech—1.53; Polish—1.41).
		≥6 METs (hour·min^−1^) (VPA): Boys (Czech—0.57; Polish—0.35); Girls (Czech—0.31; Polish—0.22).
Hubáčková et al. [55]	To assess differences in PA among primary (PS) and secondary school (SS) boys and girls in specific segments of a school day	**Energy expenditure in physical activity (kcal/kg/hour)**
		PS group: 0.50 (mdn) energy expended by boys and 0.41 by girls.
		SS group: 0.26 (mdn) energy expended by boys and 0.38 by girls.
		**Step counts (steps/hour)**
		PS group: 763 (average) steps taken by boys and 614 of girls.
		SS group: 484 (average) steps taken by boys and 554 by girls.
Mitáš et al. [51]	To identify the trends in the achievement of physical activity guidelines by Czech adolescents through objective and subjective PA monitoring	**Physical activity (MET-min/week) in girls and boys in 2010–2017**
		School: Boys (H = 7.18, *p* = 0.066, η2 = 0.005) and girls (H = 5.49, *p* = 0.139, η2 = 0.002).
		VPA: Boys (H = 9.99, *p* = 0.019, η2 = 0.009) and girls (H = 9.38, *p* = 0.025, η2 = 0.006).
		MPA: Boys (H = 10.71, *p* = 0.013, η2 = 0.010) and girls (H = 6.080, *p* = 0.109, η2 = 0.003).
Byun et al. [45]	To evaluate the feasibility and the effectiveness intervention of an PA monitoring system to promote PA in preschoolers	**SED (average)**
		GI: 31.6 min/h.
		GC: 33.6 min/h.
		**TPA (average)**
		GI: 28.4 min/h.
		GC: 26.4 min/h.
Saint-Maurice et al. [46]	To describe age, sex and season patterns in children’s physical activity behaviors during discrete time periods, both in school and at home	Recess time: 65.0% (16.5 ± 9.2 min) in MVPA. PE: 31.4% (13.9 ± 11.1 min) in MVPA.
Sanders et al. [53]	To compare PA in the physical education lessons and leisure time and compare the effect of varying accelerometer epoch length on estimates of MVPA, vigorous PA (VPA), moderate PA (MPA) and light PA (LPA)	**Time spent performing physical activity on average during Physical Education lesson:**
		MVPA—36.0612.3%; VPA—15.265.8%; MPA—20.768.2%; LPA—33.168.8% LPA—30.9614.3%; sedentary time—30.9614.3%.
		**Time spend performing physical activity on average during Leisure time:**
		MVPA—6.362.0%; VPA—1.461.0%; MPA—4.961.4%; LPA—11.462.8%, sedentary behavior—82.364.1%.
Scruggs et al. [47]	To determine a pedometer steps per minute pattern and to quantify the time students spent performing MVPA during physical education class	**Validation Sample**
		Steps*min^−1^: Boys and girls (62.04 and 64.42, respectively) and 1st grade (from 6 to 10 years) and 2nd grade students (from 10 to 14 years) (63.13 and 63.14, respectively).
		%MVPA: Boys and girls (34.56 and 35.21, respectively) and 1st grade and 2nd grade students (34.70 and 35.04, respectively).
		%VPA: Boys and girls (16.14 and 16.63, respectively) and 1st grade and 2nd grade students (17.28 and 15.39, respectively).
		**Cross-Validation**
		Steps*min^−1^: Boys and girls (63.30 and 64.15, respectively) and 1st grade and 2nd grade students (63.47 and 64.18, respectively).
		%MVPA: Boys and girls (35.22 and 34.70, respectively) and 1st grade and 2nd grade students (34.71 and 35.31, respectively).
		%VPA: Boys and girls (17.67 and 16.42, respectively) and 1st grade and 2nd grade students (17.46 and 16.25, respectively).
Mooses et al. [58]	To assess the validity of Fitbit Zip step count, MVPA and sedentary minutes	**PA monitored by Fitbip Zip and ActiGraph GT3x (average)**
		Steps (Fitbit Zip/ActiGraph GT3x): PE—2354.0 monitored by Fitbit Zip and 2008.7 ActiGraph GT3x; Recess—472.2 monitored by Fitbit Zip; 388.5 ActiGraph GT3x, r = 0.96, *p* < 0.001.
		MVPA (min) (Fitbit Zip/ActiGraph GT3x): PE—17.8 monitored by Fitbit Zip and 15.4 ActiGraph GT3x; Recess—2.1 monitored by Fitbit Zip; 2.4 ActiGraph GT3x, r = 0.56–0.72, *p* < 0.001.
		Sedentary time (min) (Fitbit Zip/ActiGraph GT3x): PE—11.1 monitored by Fitbit Zip and 13.7 ActiGraph GT3x; Recess—5.5 monitored by Fitbit Zip; 5.4 ActiGraph GT3x, r = 0.85–0.87, *p* < 0.001.
Kerr et al. [57]	To assess how PE contributes to sedentary behavior and the intensity profile of physical activity accumulated on PE days compared to those on non-PE days	**Physical activity during the PE lesson**
		On average, girls spent a larger amount of time than boys did engaged in light physical activity (7.64, *p* < 0.01, d = 1.21). In contrast, boys spent more time performing hard (6.02, *p* < 0.01, d = −1.04) and very hard (5.12, *p* < 0.01, d = −0.82) physical activity, respectively, compared with that of girls.
Janssen et al. [63]	To evaluate the effectiveness of the playground program *PLAYgrounds* on increasing PA	**Counts/min (average)**
		GI: 3924 (> 6 METs in VPA).
		GC: 2178 (3–6 METs in MPA).
		**Energy expenditure (kcal/kg/min) (average)**
		GI: 0.105 (6 METs in MPA).
		GC: 0.074 (4 METs in LPA).
		**MVPA (%)**
		GI: 77.3.
		GC: 38.7.
Saint-Maurice et al. [48]	To evaluate the utility of a multi-method approach (accelerometers plus direct observation) to better understand youth PA during recess	**MVPA**
		Boys (40.9) vs. girls (31.1): (F (1.184) = 32.22, *p* < 0.001).
		School 2 (43.5) vs. School 1 (29.2): (F (1.184) = 63.59, *p* < 0.001.
		School 2: Boys vs. girls (Meandiff = 17.9, *p* < 0.001).
		School 1: Girls vs. boys (Meandiff = 4.1, *p* < 0.001).
Hartwig et al. [54]	To develop and validate a system capable of providing feedback on PE lesson MVPA	**Physical activity PE lessons**
		Training Sample (average): Steps (2950), Steps*min^−1^ (43.5) and %MVPA (23.8).
		Validation Sample (average): Steps (3025), Steps*min^−1^ (44.9) and %MVPA (24.2).
		Convergent Sample (average): Steps (2136), Steps*min^−1^ (45.4) and %MVPA (24.2).
Sigmund et al. [52]	To identify the changes in children’s PA upon entry to first year at school and to identify the days of the school week when the students exhibit low PA values	**PA in general**
		AEE: 268.08, *p* < 0.0001, d = 1.5646.
		STEPS: 241.12, *p* < 0.0001, d = 1.3231.
		**PA of boys and girls**
		AEE (kcal/kg/day): boys (3.15, *p* < 0.0001, d = 1.7803); girls
		(2.75, *p* < 0.0001, d = 1.3501).
		STEPS (average/day): boys (2824.5, *p* < 0.0001, d = 1.5149); girls (2318, *p* < 0.0001, d = 1.2217).
Gao et al. [49]	To evaluate of the percentages of students who are overweight and obese based on the BMI and students’ physical activity level in physical education as measured by accelerometers	**Percentage of time spent performing physical activity (average)**
		Percent time spent sedentary (accelerometer; %): OW/obese (13.54) vs. Normal weight (7.64), (F (1, 146) = 10.04, *p* = 0.002, η2 = 0.06).
		Percent time performing MVPA (accelerometer; %): Normal weight (68.17) vs. OW/obese (61.14), (F (1146) = 4.89, *p* = 0.029, η2 = 0.03).
Mota et al. [59]	To observe participation in MVPA during school recess periods and to determine the relative importance of physical activity during recesses to overall daily physical activity	Daily accelerometer (counts × min^−1^): average 542 for boys and 479 for girls.
		Recess time accelerometer (counts × min^−1^): average 914 for boys and 1154 for girls.
		MVPA (min × min^−1^): averages of 142 for boys and 137 for girls.
		Recess time MVPA (min): averages of 9.2 for boys and 11.4 for girls.
Fairclough et al. [56]	Assess the day-to-day variability of children’s weekday physical activity for the whole day and when it has been segmented into discrete periods of the day	**PA to boys and girls**
		MVPA (min): Boys (32.8) and Girls (25.4).
		Intra-class correlation coefficient (95% confidence intervals) of 1 day: Boys (0.367) and Girls (0.438).
		Intra-class correlation coefficient (95% confidence intervals) of 4 days: Boys (0.698) and Girls (0.757).
Nilsen et al. [60]	Distribution of PA and SED, in particular MVPA, during preschool hours vs. time out of school	**PA of boys and girls (Average)**
		TPA (cpm): Boys (867) and girls (776).
		SED (min/day): Boys (172) and girls (186).
		LPA (min/day): Boys (186) and girls (181).
		MVPA (min/day): Boys (47) and girls (39).
Rush et al. [61]	Identify, in the context of the school day, whether a pedometer is a more effective tool, compared to an accelerometer in identifing children with low physical activity levels	**Total Steps (300 min of an average school day)**
		Total accelerometer counts (cpm): Boys—8103; Girls—6963.
		Sedentary, min/day <100 cpm: Boys—143.38; Girls—138.9.
		Light, min/day ≥100 < 1500 cpm: Boys—89.16; Girls—94.75.
		Moderate, min/day ≥1500 < 6500 cpm: Boys—64.78; Girls—63.39.
		Vigorous, min/day ≥6500 cpm: Boys—3.68; Girls—3.95.
Blaes et al. [42]	Investigate changes in time spent performing light (LPA), moderate (MPA), vigorous (VPA) and very high physical activity (VHPA) from childhood to adolescence	**School level**
		LPA (PS vs. JHS): +*p* < 0.05 (124 min per day).
		MPA (Ps vs. PS): +*p* < 0.05 (115 min per day) and MPA (PS vs. JHS): −*p* < 0.05 (233 min per day).
		VPA and VHPA (PS vs. Ps): +*p* < 0.05 (26 min per day and 25 min per day, respectively) and VPA and VHPA (PS vs. JHS): +*p* < 0.05 (26 min per day and 24 min per day, respectively).
		MVHPA (PS vs. JHS): −*p* < 0.05 (223 min per day).
Raustorp et al. [62]	To advance our knowledge of the contribution of a typical physical education (PE) class to children’s daily physical activity	**PE class**
		MVPA: 50.4% (52.5% boys and 48.3% girls).
		Total average step: 74 steps/min.

PA: Physical Activity; LPA: Light Physical Activity; VPA: Vigorous Physical Activity; AA: Academic Achievement; PS: Primary School; SS: Secondary School; MVPA: Moderate–vigorous Physical Activity; SED: Sedentary Time; GI: Group Intervention; GC: Group Control; PE: Physical Education; TPA: Total Physical Activity; VHPA: Very Light Physical Activity; MVHPA: Moderate-to-vigorous Physical Activity; JHS: Junior High Schools; Ps: Preschoolers; AEE: Activity Energy Expenditure; STEPS: Steps4. Discussion.

## Data Availability

Not applicable.

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
