# Peer review of "The Use of Wearable Technologies in the Assessment of Physical Activity in Preschool- and School-Age Youth: Systematic Review"

_ijerph, 2023, doi:10.3390/ijerph20043402_

Round 1

Reviewer 1 Report

Sousa et al reported an interesting reviews about the topic of the use of wearable devices (accelerometers and pedometers) to monitor the physical activity of childrens at school and playground. A correct monitoring of children physically activity is essential to avoid health disease and improve the quality of life. Authors summarized in readable and comprehensive way the results of several scientific articles about this topic. I suggest the publishing of this review after minor revisions.

Comments:

1) Figure 1 is unfocused. Please insert a better quality image

2) Authors insert in the text the caption Figure3 but they don't insert the figure. 

3) In the line 159 of the text the sentence obese children is repeated twice. Please correct this.

Author Response

Dear authors,

First of all, I would like to thank you for any relevant comments/suggestions for improving our article. Hopefully the corrections made will meet what was intended. Attached are the manuscript and responses to the comments made.

Best Regards

Reviewer 2 Report

The advantages of the article are:

- the presented problem concerns the physical activity of children and school youth;

- application of the secondary data analysis method, i.e. research results published by other authors. This avoids additional research costs on the same topic;

The disadvantages of the article are:

- a detailed and extensive description of studies carried out by other authors (section 3.1);

- unnecessary detailed technical and technological characteristics of devices used to measure physical activity (Table 2).

- the title of the article indicates that a group of children and adolescents of school age is being studied. The article also presents the results of physical activity tests of children aged 3, 4, and 5 years. They are not school age children.

I suggest supplementing the applications with suggestions on what parents and teachers can do at school to increase the physical activity of children and adolescents at school age.

Author Response

Dear authors,

First of all, I would like to thank you for any relevant comments/suggestions for improving our article. Hopefully the corrections made will meet what was intended. Attached are the responses to the authors.

Best Regards

Reviewer 3 Report

In the search methodology, it would be advisable for the authors to specify the date criteria used for the search.

In the methods, in line 59th, the autors remark: “In addition, relevant research articles published between January 2000 and August 2022 were collected”.

It would be convenient for them to specify what they refer to, what was the exact methodology they used to add these articles.

In Table 1 the authors analyze the Sample size, Device, Results…

It would be advisable for the authors to make some mention of these results apart from putting them in the table in the text. And it is only done by the sample size. Similarly occurs in Table 2.

The authors refer to a figure 3, but do not include it even though the caption text appears.

I would try to synthesize more information from the 3 tables to make it easier for the reader to read the article.

In the discussion the authors discuss results that they do not explain in the results part. For this reason, I consider it would be advisable to either add it to the results or rewrite the discussion paragraph. In this section it would also be advisable to mention the limitations of the study.

In the case of the conclusions, I think the authors could add and emphasize why this article is relevant and what conclusions they reached to improve knowledge of the field, not just that more studies are needed on the subject.

Author Response

(The authors gave the same response as above.)

Reviewer 4 Report

The topic is important and of interest. I offered both broad and some particular comments.  This is a well-written manuscript, and/but requires “moderate” revision. Thank you.

Abstract: Please check the length, perhaps this abstract is too short, and appears incomplete particularly for the last sentence. Please include further information about your primary findings, as well as any relevant suggestions.

In the introduction, you may need to highlight the current issues related to physical activity among children or adolescent, declining or something. Then relate to how much wearable tech can influence positive changes in physical activity.

In the introduction, please provide definition and explain more about wearable technologies – in one paragraph. What is it for? Why is it so popular nowadays? Advantages? Different types? (Device) How does it works?

Line 139: 1st grade is about what age? Please add to facilitate global readership (and consider all others).

In the methods, a systematic search attempts to identify all studies. Ok, have all the database searches been done exhaustively?  I wish to highlight that this is an iterative process and the data searches will need to be undertaken “several/many times” with various combinations of search terms, and across the suite of databases. 

In the discussion section, I think, the authors need to highlight the major findings of the study in the first paragraph of discussion section. These points can be used to set out the next paragraphs (one finding one paragraph) where the authors contrast/compare the findings of the review to existing literature.

After reading the conclusion, the reader should understand why your study is important to them. A conclusion is a synthesis of essential arguments, not just a rehash of your findings. Suggest improve its clarity.

Line 269: Please add more information or explain more: “to evaluate the physical condition”

Line 270: Please add examples in bracket.

Additionally, can you also please highlight some information regarding the reliability and validity of pedometers and accelerometer?

Author Response

(The authors gave the same response as above.)

Round 2

Reviewer 3 Report

Congratulations. The article has improved substantially.